# Genetic and Pathogenic Analysis of a Novel Porcine Epidemic Diarrhea Virus Strain Isolated in the Republic of Korea

**DOI:** 10.3390/v16071108

**Published:** 2024-07-10

**Authors:** Dae-Min Kim, Sung-Hyun Moon, Seung-Chai Kim, Ho-Seong Cho, Dongseob Tark

**Affiliations:** 1Laboratory for Infectious Disease Prevention, Korea Zoonosis Research Institute, Jeonbuk National University, Iksan 54531, Republic of Korea; daeminkk@gmail.com; 2College of Veterinary Medicine, Bio-Safety Research Institute, Jeonbuk National University, Iksan 54596, Republic of Korea; chunsu17@naver.com (S.-H.M.); leesor2@jbnu.ac.kr (S.-C.K.); hscho@jbnu.ac.kr (H.-S.C.)

**Keywords:** porcine epidemic diarrhea virus, novel strain, characterization

## Abstract

Porcine epidemic diarrhea (PED), caused by the porcine epidemic diarrhea virus (PEDV), emerges annually in several Asian countries. Its major symptoms include watery diarrhea, vomiting, anorexia, and dehydration. PED outbreaks incur significant economic losses. The efficacy of vaccines is limited by viral mutations and insufficient intestinal mucosal immunity. Therefore, new vaccines against these recent variants are urgently needed. Herein, we isolated and genetically characterized a novel Korean PEDV strain using NGS. Comparative genomic analysis demonstrated that the CKK1-1 strain belonged to genogroup 2. The isolated strain was cultured in sodium-glycochenodeoxycholic acid for 180 passages. Typically, PEDV isolation and passage require proteases, such as trypsin. However, the CKK1-1 strain adapted to this atypical culture condition, achieving a high titer of 8.83 ± 0.14 log TCID_50_/mL. In vitro biological analysis revealed no cell syncytium formation without trypsin; however, a cell-lysis-type cytopathic effect was noted. Notably, pathogenicity evaluation showed that CKK1-1 p0 exhibited naturally weakened virulence in five-day-old piglets, while piglets administered with CKK1-1 p180 exhibited 100% survival and reduced clinical symptoms. Collectively, our data demonstrate that this Korean PEDV strain, attenuated through atypical culture conditions with Na-glycochenodeoxycholic acid, has potential as a vaccine candidate, providing valuable insights into the genetic variation in and pathogenicity of PEDV.

## 1. Introduction

Porcine epidemic diarrhea (PED), caused by the porcine epidemic diarrhea virus (PEDV), is a highly contagious enteric disease which can develop in pigs of all ages, with neonatal piglets being particularly vulnerable to fatal infections. Its major symptoms include watery diarrhea, vomiting, anorexia, and severe dehydration. PED outbreaks have led to considerable losses in the livestock industry. PEDV was first discovered in England in 1971 and was subsequently identified in Belgium in 1978 [1]. The PED outbreak initially spread locally in Europe but has since spread to most Asian countries, including the Republic of Korea, Japan, Thailand, and China. In October 2010, a highly pathogenic variant emerged in China and spread rapidly worldwide. This variant caused significant losses in the United States in 2013, after which it spread to Mexico and Canada. Since the discovery of this variant in the United States and China, outbreaks of highly pathogenic PED strains have occurred in Asian countries. In the Republic of Korea, PED first emerged in 1992 and has since been reported annually [2,3]. Moreover, recent reports have suggested that the variant strain is circulating in fields in the Republic of Korea.

PEDV is a single-stranded, positive-sense RNA virus belonging to the family *Coronaviridae* in the order *Nidovirales* [4]. The viral genome, approximately 28 kb in size, is composed of a 5′ untranslated region (UTR), a 3′ UTR, and at least seven open reading frames (ORF 1a, ORF 1b, and ORFs 2–6). The ORFs encode non-structural proteins (ORFs 1a, 1b), structural proteins (ORFs 2, 4–6), and accessory proteins (ORF 3). Among the ORFs, ORFs 1a and 1b encode two replicase polyproteins, pp1a and pp1b, which are proteolytically processed by viral proteases to produce 16 non-structural proteins (NSPs 1–16). These NSPs facilitate viral replication, transcription, and translation. Recent studies have reported that NSPs play a role in the viral life cycle, acting as interferon antagonists during viral infection [5]. The remaining ORFs encode four structural proteins: spike (S), envelope (E), membrane (M), nucleocapsid (N), and accessory protein ORF 3. The spike protein is a type I glycoprotein that forms peplomers on the virion, playing a critical role in membrane fusion after the virus attaches to the cell receptor [6]. In addition, the spike protein contains major neutralizing epitopes such as COE (499–638 aa), SS2 (748–755 aa), and SS6 (746–771 aa), which are important for investigating genetic relationships due to their sequence diversity [7]. Envelope proteins aid in viral assembly and budding. Highly conserved membrane proteins are involved in envelope assembly. This multifunctional nucleocapsid protein participates in viral genome replication, virion assembly, and pathogenicity. The genome organization is as follows: 5′ UTR-ORF1a/1b-S-ORF3-E-M-N-3′ UTR. Based on genetic analysis, PEDV strains are predominantly divided into two groups: classical or recombinant and low-pathogenicity (genogroups 1 and G1), and field-epidemic or pandemic and high-pathogenicity (genogroups 2 and G2). Each group was further classified into two sub-genogroups: G1a, G1b, G2a, and G2b [8].

Vaccination is recognized as the best method to control the spread of PEDV. Currently, commercial PED vaccines fall into two categories: live-attenuated and inactivated vaccines. Live-attenuated vaccines (LAV), produced through the modification of a wild virus, are commonly used to prevent PED outbreaks. However, the efficacy of LAVs is controversial because of the antigenic and genetic differences between the vaccine and circulating strains. Traditionally, virulent strains (G1), such as SM98 and DR-13, have been attenuated using in vitro culture systems for use as live or inactivated vaccines in the Republic of Korea. According to several vaccination studies, the survival rate of piglets has improved with the use of various commercial vaccines and multiple vaccination programs. [9]. However, despite the use of available vaccines, the Republic of Korea has experienced reemerging PED outbreaks (G2-b), indicating that commercial vaccines do not provide significant protection against recent strains. To resolve this issue, there is a need to develop more effective and safer vaccines against new variant strains. Additionally, the latest circulating strains need to be tracked through a comparative analysis of viral genome sequences. In this study, a prevalent strain, CKK1-1, was isolated from a farm in the Republic of Korea, and its genome was analyzed using next-generation sequencing (NGS). The sequence was compared to that of a reference strain to understand its genetic relationship. Subsequently, the virus was serially passaged in a cell culture system containing Na-glycochenodeoxycholic acid (Na-GCDCA). Growth kinetics and biological characteristics were determined. Additionally, an animal experiment was conducted to evaluate the pathogenicity of these strains in 5-day-old piglets. Finally, the whole-genome sequences of the CKK1-1 parental and modified strains were compared to investigate the associated genetic alterations. In this study, a recently prevalent PEDV strain was attenuated for virulence, providing an important basis for the preparation of live-attenuated vaccine (LAV) candidates.

## 2. Materials and Methods

### 2.1. Cell Culture

Vero cells were cultured in Dulbecco’s Modified Eagle Medium (DMEM; Gibco, Grand Island, NY, USA) with 5% fetal bovine serum (FBS, Gibco, USA) and antibiotics (100 U/mL penicillin, 100 ug/mL streptomycin, Gibco, USA) at 37 °C in a humidified incubator with 5% CO_2_.

### 2.2. Isolation of Porcine Epidemic Diarrhea Virus

The small intestines of piglets infected with porcine epidemic diarrhea virus were collected from a PED-affected farm in the Republic of Korea in November 2019. The intestines were homogenized with sea sand and virus growth media containing DMEM supplemented with antibiotics, 0.02% yeast extract (Difco, Detroit, MI, USA), 0.3% tryptose phosphate broth (TPB, Difco, USA), 1.5 µg/mL L-1-tosylamide-2-phenylethyl chloromethyl ketone (TPCK)-treated trypsin (Sigma, Chicago, IL, USA), and 100 µM/mL Na-glycochenodeoxycholic acid (Na-GCDCA, Sigma, USA) to produce a 10% tissue homogenate solution. After centrifugation at 3000 rpm for 20 min, the supernatant was filtered using a 0.22 µm syringe filter (Millipore, Burlington, MA, USA). Subsequently, the supernatant was inoculated into monolayers of Vero cells. The cells were monitored daily under a microscope (Leica Microsystems, Wetzlar, Germany) until distinct morphological alterations were observed. PEDV-infected cells were then blindly passaged to confirm obvious cytopathic effects (CPEs), such as syncytium formation.

### 2.3. Development of PEDV Strain CKK1-1

The isolated virus was serially cultured for 180 passages in viral growth media. Vero cells were seeded at a density of 2 × 10^6^ in T-25 flasks. The monolayered cells were washed three times with phosphate-buffered saline (PBS) and then inoculated with diluted virus stock at a multiplicity of infection (MOI) of 0.1. After 1 h, the inoculated virus was removed, and fresh viral growth medium was added to the cells. The cells were monitored daily until an 80% cytopathic effect was observed. The PEDV-infected cells were then stored at −80 °C for the next passage.

The isolated PEDV strain was passaged for 180 generations under various conditions (Table 1). The virus was initially passaged using virus growth media, and the concentration of TPCK-treated trypsin was gradually reduced to selectively propagate growth-competent viruses using Na-GCDCA. Initially, the PEDV strain was passaged in virus growth media containing 1.5 μg/mL TPCK-treated trypsin for up to 27 generations. Subsequently, the trypsin concentration was reduced to 1 μg/mL for 40 generations. Following this, the CKK1-1 strain underwent ten passages with 0.5 μg/mL TPCK-treated trypsin, after which the viruses were adapted to 0.2 μg/mL TPCK-treated trypsin. After four passages with 0.2 μg/mL TPCK-treated trypsin, the CKK1-1 strain was grown in media containing only Na-GCDCA. Finally, the CKK1-1 strain was passaged exclusively with Na-GCDCA for 99 generations, totaling 180 passages.

### 2.4. Titration of Porcine Epidemic Diarrhea Virus

Titration of the PEDV was performed using the Reed–Muench method [10]. Vero cells were seeded in a 96-well plate at a density of approximately 4 × 10^4^ cells/well. The cells were then inoculated with 10-fold serially diluted viral stocks. After 1 h of incubation, the inoculated virus was removed and replaced with fresh viral growth medium. The cells were monitored daily by microscopy for up to 5 days post-inoculation to observe cytopathic effects (CPEs).

### 2.5. Immunofluorescence Assay

Vero cells were infected with the virus at a multiplicity of infection (MOI) of 0.1. After incubating the cells for 24 h, the cells were fixed with 80% cold acetone for 15 min. The fixed cells were stained with an anti-PEDV monoclonal antibody (Median Diagnostic, Chuncheon-si, Republic of Korea) for 1 h at room temperature. Subsequently, the cells were treated with an anti-mouse secondary antibody conjugated to Alexa Fluor 488 (Cell Signaling Technology, Danvers, MA, USA) for 1 h at room temperature in the dark. Following incubation, the cell nuclei were counterstained with 4′,6-diamidino-2-phenylindole (DAPI, Sigma, USA). The stained cells were visualized using a CELENA^®^ S digital imaging system (Logos Biosystems, Anyang, Republic of Korea).

### 2.6. Evaluation of Pathogenicity, CKK1-1

Twelve 5-day-old piglets were obtained from a PED non-outbreak farm and randomly divided into three experimental groups in separate rooms. All piglets were fed a commercial milk substitute three times daily during the study period. In the positive group, the piglets (*n* = 4) were orally inoculated with a 10% tissue homogenate solution (CKK1-1 passage 0), while the negative control group (*n* = 4) received cell culture media. Additionally, the remaining piglets (*n* = 4) were inoculated with passaged PEDV (passage no. 180) and designated as the CKK1-1 p180 group. After inoculation, all piglets were monitored daily for clinical symptoms, and rectal swabs were collected until the end of the experiment. The clinical scores for fecal consistency were as follows: 0, solid; 1, pasty; 2, semi-liquid; 3, liquid; and 4, death [11]. The piglets were examined for histopathological lesions at the end of the experiment after euthanasia.

### 2.7. Quantitative Real-Time PCR

Viral shedding in rectal swab samples was quantified using quantitative real-time PCR (qRT-PCR). Rectal swab samples were collected daily in a clinical viral transport medium (CTM, Noble-bio, Suwon-si, Republic of Korea). The viral genome was extracted from these samples using a QIAamp Viral RNA Mini Kit (Qiagen, Hamburg, Germany). RNA was quantified by qRT-PCR using Cellscript^TM^ RT-Q Green Blue Master Mix (Cell-safe, Daejeon, Republic of Korea) following the manufacturer’s instructions. The viral copy number was quantified using a standard curve generated with PEDV M gene-specific primer sets (forward primer: 5′-GGTTCTATTCCCGTTGATGAGGT-3′, reverse primer: 5′-AACACAAGAGGCCAAAGTATCCAT-3′), as described previously [12]. Viral genome amplification was performed using CFX96 real-time PCR (Bio-Rad Laboratories, CA, USA).

### 2.8. Histopathology and Immunochemistry

For the analysis of histopathological lesions, small-intestine tissues were collected, fixed in 10% formalin for 24 h at room temperature after euthanasia, and embedded in paraffin. The formalin-fixed paraffin-embedded tissue blocks were cut into 4 µm thick sections using a microtome (Thermo Fisher, USA). The tissue sections were deparaffinized with xylene for 10 min at room temperature, followed by treatment with decreasing concentrations of ethanol (100%, 90%, 70%, and 50%, respectively) for 5 min each. After deparaffinization, the tissue sections were stained with hematoxylin and eosin. Alternatively, some sections underwent antigen retrieval using citrate buffer (pH 6.0) at 95 °C for 30 min, followed by cooling at room temperature for 20 min. Subsequently, the sections were incubated with a PEDV-specific antibody (Median Diagnostic, Republic of Korea) overnight at 4 °C and then treated with horseradish-peroxidase-conjugated anti-mouse immunoglobulin G (IgG) antibody (Vector Laboratories, Newark, CA, USA) at room temperature for 1 h. For visualization, the sections were covered with 3,3′-diaminobenzidine (DAB, Vector Laboratories, USA) and counterstained with methyl green. The stained slides were imaged using a BX53 microscope and DP80 camera (Olympus, Tokyo, Japan).

### 2.9. Next-Generation Sequencing

The viral whole-genome sequence was determined using next-generation sequencing (NGS) technology. Briefly, the viral genome was extracted using a QIAamp Viral RNA Mini Kit (Qiagen, Germany). Reverse transcription and library preparation were performed using the target enrichment method, as described previously [13].

### 2.10. Sequence Analysis and Phylogenetic Tree

The reference genome sequence was obtained by pooling data from GenBank for multiple alignments. Multiple sequence alignments were performed using the Clustal Omega program. Subsequently, the aligned sequences were analyzed using BLASTn to compare nucleotide identity percentages, and a phylogenetic tree was constructed using the MEGA X software (version 11) [14]. The genetic evolutionary distance model was computed using the Tamura–Nei method. Additionally, a phylogenetic tree was constructed using the neighbor-joining method, and the percentage of reliability values for each node was determined through bootstrap analysis with 1000 replicates. Deduced spike protein alignment was finally performed using MULTALIN (http://multalin.toulouse.inra.fr/multalin/multalin.html (accessed on 28 March 2000)) [15].

## 3. Results

### 3.1. Genetic Characterization of PEDV Strain CKK1-1

The sequence arrangement indicated that the complete-genome sequence of strain CKK1-1 (GenBank accession no. OM714830) was 28,037 nucleotides (nt) in length, excluding the 3′-end poly(A) tail. The genome positions were as follows: open reading frame 1a/b (ORF 1a/b), nt 292 to 20,636; spike gene, nt 20,633 to 24,793; ORF3, nt 24,793 to 25,467; envelope gene, nt 25,488 to 25,678; membrane gene, nt 25,686 to 26,366; and nucleocapsid gene, nt 26,378 to 27,703. We compared the genetic identity between strain CKK1-1 and the reference strains from the whole genome to other regions (Table 2). The nucleotide identity of the entire genome of all the PEDV strains analyzed ranged from 96.23% to 99.27%. Specifically, the genogroup 1 strains (including CV777, CH/S, SM98, DR13_virulent, DR13_att, LZC, JS2008, and SD-M) exhibited lower identity, with identities ranging from 96.31% to 97.37%. In contrast, the genogroup 2 strains shared nucleotide identities of 97.89% or higher. The highest nucleotide identity observed across the entire genome was 99.27%, found in the TC_PC177-P2 strain. Furthermore, the genetic identity of various genomic regions was confirmed by the following percentages: 5′ UTR (94.94% to 100%), ORF1a (96.68% to 99.76%), ORF1b (97.38% to 99.57%), spike (92.70% to 98.98%), ORF3 (94.84% to 99.55%), envelope (95.44% to 99.56%), membrane (96.85% to 99.71%), nucleocapsid (95.35% to 99.63%), and 3′ UTR (96.59% to 99.10%). Among the ORFs, the highest identity variation occurred within the spike gene. In the identity of the spike gene, USA/Colorado/2013 showed the highest identity with CKK1-1 at 98.98%, and LZC showed the lowest identity (92.70%). Furthermore, we compared the deduced amino acid sequences of CKK1-1 with the reference sequence. The amino acid identity comparisons were as follows: ORF1a (93.02% to 99.58%), ORF1b (98.80% to 99.77%), spike (92.07% to 98.47%), ORF3 (96.36% to 100%), envelope (94.59% to 100%), membrane (96.38% to 99.55%), and nucleocapsid (96.54% to 100%). As expected, and consistent with the nucleotide data, the spike protein exhibited the highest variability in the deduced amino acid mutations. To confirm genetic alterations in the neutralizing epitopes in the spike protein, we compared the alignment of the deduced spike protein between CKK1-1 and other strains (Appendix A). Mutations were identified in the N-terminal domain (NTD) of the deduced spike protein, with the CKK1-1 strain exhibiting four unique substitutions (T24A, R/D86K, I158V, and M/I220T) compared to the other strains. Subsequently, we examined mutations in the neutralizing epitope regions (COE, SS2, SS6, and 2C10). Interestingly, the neutralizing epitopes (COE, SS2, SS6, and 2C10) showed no distinct alterations among the strains. However, the CKK1-1 strain exhibited seven substitutions outside the neutralizing epitope regions (S928T, N1036S, T1188I, T/M1207I, E136D, Q/R1308K, and I1343V). Based on a comparative analysis of the sequence, we confirmed that the CKK1-1 strain was a genetically distinct strain that has not been reported so far, without alterations in the neutralizing epitope regions.

### 3.2. Phylogenetic Analysis

After determining the whole-genome sequence using next-generation sequencing, we generated a phylogenetic tree to understand the relationship between CKK1-1 and the reference strains. Phylogenetic trees were constructed based on the whole genome as well as individually based on the ORF1a, ORF1b, spike, ORF3, envelope, membrane, and nucleocapsid genes. Forty reference strains were used for clustering (Figure 1). The trees revealed that these strains could be classified as genogroups 1 and 2. Genogroup 1 is the classic strain, with lower pathogenicity, whereas genogroup 2 is pandemic-causing and highly pathogenic. In the whole-genome analysis, the CKK1-1 strain was grouped with genogroup 2 and closely clustered with the OH851 strain in an adjacent clade. Moreover, the CKK1-1 strain was classified within genogroup 2 in trees based on other ORFs, indicating a distinct phylogenetic clade. However, the classical strains (CV777, SM98, and DR13) circulating in the Republic of Korea belonged to genogroup 1. Typically, phylogenetic trees of porcine epidemic diarrhea viruses have been constructed based on the spike gene. In the spike gene tree, the CKK1-1 strain was closely associated with the JSCZ1601 strain in a neighboring clade. Altogether, CKK1-1 was affiliated with a new clade within genogroup 2 when compared to the reference strains.

### 3.3. Virus Isolation and Biological Characteristics

We obtained PEDV-positive small-intestine samples from a PED outbreak farm in the Republic of Korea. After genome analysis, we successfully isolated PEDV in Vero cells. After three blind passages, the PEDV-infected cells exhibited typical cytopathic effects (CPEs), such as cell fusion, syncytium formation, and clumping. Subsequently, the CPEs became more pronounced with each passage. The CKK1-1 strain was passaged according to laboratory procedures (Table 1). Interestingly, CKK1-1 displayed atypical CPEs, such as cell lysis and detachment (Figure 2A). As the CKK1-1 strain was continually passaged with decreasing concentrations of TPCK-treated trypsin, the nature of the CPEs changed. Initially, the CPEs appeared within 72 h post-inoculation at early passage numbers; however, at late passage numbers, major CPEs were observed within 20 h post-inoculation. Furthermore, we confirmed the dependence of the CPE characteristics on the passage number using an immunofluorescence assay. PEDV was detected in the cytoplasm as green signals, whereas no signals were observed in the controls. Remarkably, the CKK1-1 strain from the early passage numbers exhibited signals only with syncytium formation. Conversely, the CKK1-1 strain with 180 passages displayed signals from the adjacent cytoplasm in each cell type (Figure 2B). In summary, we successfully isolated a new PEDV strain that is circulating in the Republic of Korea and cultivated it under various conditions. Moreover, we verified that the CPE characteristics varied according to the cultivation conditions.

### 3.4. Growth Kinetics of CKK1-1

We measured the virus growth kinetics and titrated the CKK1-1 strain every 10 passages (Figure 3). At the 10th passage, CKK1-1 was passaged using virus growth media, resulting in a titer of 4.00 ± 0.25 log TCID_50_/mL. The CKK1-1 titer gradually increased with each passage. After 40 passages, the titer was consistently maintained above 7.0 log TCID_50_/mL, reaching its peak at 8.83 ± 0.14 log TCID_50_/mL at passage number 130. Furthermore, the CKK1-1 strain grown with only Na-GCDCA tended to achieve higher titers than those grown with TPCK-treated trypsin (passage number > 80 generations). In conclusion, we determined that the CKK1-1 strain was successfully cultivated using only Na-GCDCA and that high titers were sustained.

### 3.5. Evaluation of the Pathogenicity of CKK1-1

The CKK1-1 strain, a new strain isolated from the Republic of Korea and cultivated with bile salts in viral growth media, exhibited robust growth with high viral titers. Consequently, we investigated the pathogenicity and in vivo phenotypic characteristics of this strain. The pathogenicity of the CKK1-1 strain was evaluated in 5-day-old piglets (Table 3). Twelve piglets were randomly assigned to three groups of four animals each and housed individually. The piglets in the CKK1-1 p0 group were orally inoculated with a 10% solution of small-intestine homogenate. The CKK1-1 p180 group was challenged with a CKK1-1 strain passaged for 180 generations (7.0 log TCID_50_). The remaining piglets were administered cell culture medium to serve as the negative control group. Clinical symptoms were monitored daily, and fecal swabs were collected to determine viral shedding until the end of the animal experiments. Initially, all piglets appeared healthy with no clinical signs during the acclimation period. After oral inoculation, the incidence of severe diarrhea ranged from 0% to 100%. Specifically, the piglets in the CKK1-1 p0 group experienced severe diarrhea at 1–2 days post-inoculation, whereas the piglets in the other groups did not exhibit diarrhea. No fatalities were observed in any group (Figure 4A). The fecal consistency scores indicated a tendency toward recovery in the CKK1-1 p0 group over the course of the experiment, following the initial onset of severe diarrhea (Figure 4B). Furthermore, the detection of the viral genome in the rectal swabs began at 2 days post-inoculation, peaked at 3 days post-inoculation in the CKK1-1 p0 group, and then decreased gradually during the experiment. No viral genomes were detected in any of the other groups (Figure 4C).

### 3.6. Clinical Symptoms and Histopathological and Immunohistochemical Staining

Clinical symptoms were monitored daily following infection with either the CKK1-1 strain (p0 and p180) or medium, and all piglets underwent necropsy 7 days post-inoculation. Gross examination revealed that the CKK1-1 p0 group exhibited diarrhea around the buttocks, whereas none of the other groups exhibited such symptoms (Figure 5). Additionally, the small intestines of the CKK1-1 p0 piglets were filled with yellowish water, and the walls appeared thin and transparent. Conversely, no significant gross lesions were observed in any of the other groups. Microscopic examination revealed viral enteritis characterized by shortening, fusion, and atrophy of the villi as well as exfoliation of enterocytes in sections of the small intestine (duodenum, jejunum, and ileum) in the CKK1-1 p0 group. Interestingly, the CKK1-1 p180 group exhibited only subtle changes in the villi. Similarly, the negative control group showed no obvious gross lesions in sections of the small intestine. Subsequently, the distribution patterns of the viral antigens were analyzed using immunohistochemical staining, which revealed brown signals representing the PEDV N protein antigen. PEDV antigens were predominantly detected in the cytoplasm of the atrophic villous epithelial cells. In the CKK1-1 p0 group, the antigen was distributed throughout the small intestine. Surprisingly, although microscopic lesions were scarcely observed in the villi, the presence of viral antigens in the small intestines of the piglets was confirmed in the CKK1-1 p180 group. Conversely, PEDV antigen was not detected in the negative control group. In summary, the CKK1-1 p180 strain exhibited attenuation, as evidenced by the absence of gross lesions and microscopic changes. However, viral antigens were detected in the small intestines of the piglets in the CKK1-1 p180 group.

### 3.7. Comparison of Genetic Mutations in the CKK1-1 Strain Depending on Passage Number

To collect genetic mutation information according to the passage number of the CKK1-1 strain, we compared the alterations in the nucleotides and amino acids between the passage 0 and 180 strains (Table 4). A total of 65 nucleotide mutations were detected, including 27 mutations in the non-structural protein region (ORF 1a/1b), 37 mutations in the structural protein region (spike, envelope, membrane, and nucleocapsid), and 1 mutation in the accessory protein region (ORF 3). While five out of the twenty-seven nucleotide mutations occurred in the ORF 1a/1b region, there were no differences in amino acids across the passaged generations (44F, 1264 L, 1322N, 3817N, and 5537 L). However, the remaining twenty-two nucleotide mutations in ORF 1a/1b resulted in amino acid substitutions. Among the mutations in the structural protein region, the most variable was in the spike protein, which carried 29 mutations. Notably, the insertion of a nucleotide at the C-terminus of the spike region caused an early stop codon (1380*) in the CKK1-1_P180 strain. The most conserved region was the envelope, with a single amino acid substitution (P70L). Additionally, several genetic point mutations in other structural protein regions were found to result in amino acid substitutions in the membrane (I12V and I24L) and nucleocapsid (R166I, R184P, and G377E). Furthermore, ORF 3, a well-known accessory protein, contained an amino acid substitution (D191Y).

## 4. Discussion

Outbreaks of porcine epidemic diarrhea (PED) have been reported annually in Asian countries, including the Republic of Korea, Japan, Taiwan, and China, since its emergence, resulting in significant economic losses to the swine livestock industry [16]. The development of vaccines has been recognized as a valuable strategy to prevent the spread of this pandemic. Inactivated and live-attenuated vaccines are considered the most effective classical approaches because whole-virus vaccines possess sufficient antigenicity and immunogenicity to induce immunity. However, the use of these vaccines has several drawbacks, including the possibility of virulence relapse, the requirement for a high virus titer, and difficulty in selecting low-pathogenic strains [17]. Until recently, genogroup 1 strains were predominant in the Republic of Korea, and several vaccines derived from classical PEDV strains were developed through serial passages in cell culture systems. However, various PED outbreaks of genogroup 2 viruses have more recently occurred in the Republic of Korea. Due to genetic mutations in the viral genome, the development of new vaccines targeting circulating virulent strains has become necessary [18].

In the present study, we isolated a distinct Korean PEDV strain, designated as CKK1-1, which we passaged with bile salt for up to 180 passages. The pathogenicity of CKK1-1 was subsequently assessed in 5-day-old piglets. The CKK1-1 p180 strain showed attenuation, as evidenced by the reduction in clinical symptoms, viral shedding, gross lesions, and histopathological changes in the piglets.

PEDV comprises seven open reading frames (ORFs 1a, 1b and ORFs 2–6). ORFs 1a and 1b encode 16 nonstructural proteins (NSPs 1–16), ORF 2 and ORFs 4–6 encode structural proteins (S, E, M, and N), and ORF 3 encodes an accessory protein. Recent studies have shed light on the functions of each protein, revealing their roles not only in gene replication, transcription, and translation but also in viral invasion into host cells and immune evasion [19]. Additionally, PEDV is classified into two genogroups based on genotype: genogroup 1 (G1; classical) and genogroup 2 (G2; field-epidemic), which are further subdivided into genogroups 1a, 1b, 2a, and 2b. In particular, the G1b and G2b strains have been predominant in the Republic of Korea over the past decade. G1b emerged through recombination events between the classical G1a and epidemic G2 strains, whereas G2b, which is responsible for most domestic outbreaks, continues to evolve via genetic mutations. Given the ongoing evolutionary process of the virus, which is characterized by the accumulation of mutations and/or recombination events for its survival in the field, it is imperative that researchers conduct comparative genetic analyses to identify unidentified PEDV genotypes that may emerge locally or globally through genetic mutations or recombination [20].

To date, vaccines are regarded as the most effective method for controlling PEDV. While various vaccine platforms exist, including inactivated, live-attenuated, nucleic acid, subunit, and virus-like particle vaccines, live-attenuated vaccines using whole-virus formulations are widely recognized to elicit the highest immunogenicity [21]. Live-attenuated vaccines can be developed by passaging virulent strains in cell culture systems. In the case of PEDV, the presence of exogenous proteases is essential for the isolation and propagation of the virus in cell cultures. This protease enhances infectivity by facilitating viral entry into the cells and cell-to-cell propagation. Although the detailed mechanism of protease action is not fully understood, it is known that viral glycoproteins play a role in attaching to receptors and fusing with the cell membrane through protease-mediated proteolysis. Studies on the role of proteases in several coronaviruses have further been conducted. For example, mouse hepatitis virus (MHV) strain A59, infectious bronchitis virus (IBV), and influenza virus use furin protease to increase infectivity. Furin is responsible for cleaving other proteins into their mature or active forms and is predominantly found in the Golgi apparatus [22]. Furthermore, the spike proteins of severe acute respiratory syndrome coronavirus (SARS-CoV) and Middle East respiratory syndrome coronavirus (MERS-CoV) are activated by membrane-bound protease serine 2 (TMPRSS2). TMPRSS2 is widely distribute in the cell membranes of the epithelial cells found in the lungs, prostate, heart, liver, and gastrointestinal tract [23]. Similarly, the viral glycoprotein (GP) 1 of the Ebola virus is cleaved by the host protease cathepsin, which is present in endosomes and lysosomes [24]. Unlike other coronaviruses, most PEDVs have been shown to require the exogenous protease trypsin for the efficient propagation. Trypsin is predominantly localized in the enterocytes of the small intestine. Upon stimulation, trypsinogen, which is produced at the pancreas in its inactive form, is secreted into the small intestine in its active form [25]. Trypsin cleaves the PEDV spike protein into two subunits: the N-terminal S1 subunit, which binds to target receptors, and the C-terminal S2 subunit, which is responsible for fusion with the cell membrane [26]. Previous studies have further demonstrated the development of trypsin-independent PEDV strains through serial passaging under various culture conditions using elastase and glycochenodeoxycholic acid (GCDCA). A recent study generated live-attenuated PEDV vaccine candidates under six different culture conditions. These strains were compared and analyzed for their growth characteristics and pathogenicity, and the results indicated that the strain passaged with GCDCA exhibited a high titer and attenuation [27,28]. Generally, bile acid synthesis occurs in hepatocytes, where primary bile acids are synthesized through a series of enzyme cascades. After synthesis, some bile acids are conjugated with taurine or glycine and can be secreted into the duodenum or stored in the gallbladder. When acidic and fatty chymes stimulate enteroendocrine I cells, bile acids are secreted into the duodenum [29]. Several viruses utilize bile acids to promote infection and replication. For instance, Chang et al. [30] showed that porcine enteric calicivirus (PEC) could adapt only to the intestinal content (IC) in cell culture systems, with bile acids identified as active factors in the IC. Furthermore, human sapoviruses (HuSaVs) causing acute gastroenteritis require bile acids in vitro [31]. Although the mechanisms underlying the interaction between viruses and bile acids are not fully understood, creating an environment similar to that of the intestine is crucial for viral adaptation.

In our animal experiments, we assessed the pathogenicity of the CKK1-1 strain by dividing the groups into CKK1-1 p0, CKK1-1 p180, and a negative control. Interestingly, the piglets in the CKK1-1 p0 group did not experience mortality (mortality rate: 0%), despite severe clinical symptoms developing at 1–2 days post-inoculation. Conversely, the piglets inoculated with the CKK1-1 p180 strain did not exhibit mortality nor any clinical symptoms throughout the experiment. Furthermore, microscopic analysis comparing the clinical lesions between the CKK1-1 p0 and CKK1-1 p180 groups revealed that the CKK1-1 p180 strain exhibited weaker virulence than the CKK1-1 p0 strain. Consequently, we isolated a new virus that did not cause mortality, which was subsequently confirmed to have naturally attenuated pathogenicity. After evaluating virulence, we analyzed the genetic characteristics of the strain with reference strains, and no alterations were observed in the neutralizing epitopes among the strains. In the CKK1-1 p180 strain, deletions were identified in the C-terminus of the spike protein, a region known to play a role in endocytosis and ER retrieval signals, including the YxxΦ and KVHVQ motifs, respectively. Hou et al. [32] demonstrated that the deletion of both motifs resulted in weak virulence in piglets. Information on genetic changes will aid in securing genetic resources and identifying pathogenic regions.

Further studies are required to assess antibody levels (IgG and IgA) after sow vaccination. As sows induce immunity against PEDV and as maternal antibodies are vertically transferred to protect neonatal piglets through colostrum and milk, it is crucial to evaluate this process. Additionally, because the placenta of sows is impermeable, the induction of sIgA in the colostrum is particularly important during the neonatal period. Subsequently, we will evaluate whether vaccinated sows produce PEDV-specific antibodies (IgG and sIgA) in their blood and colostrum and whether the induced immunity provides protection against virulent PEDV challenges. To better understand the genetic characteristics of PEDV, we plan to develop an infectious cDNA clone using reverse genetics. This cDNA clone could then be used to study key virulence factors, enable rapid responses to re-emerging epidemic strains, and elucidate the mechanisms of protease effects.

Collectively, we successfully isolated a distinct PEDV strain in the Republic of Korea, designated as CKK1-1, which exhibited distinct genetic characteristics. CKK1-1 was grown under atypical culture conditions in the presence of bile acids for up to 180 passages. We further observed that the CKK1-1 strain adapted to these culture conditions, achieving a high titer. Furthermore, the biological properties of the CKK1-1 strain were evident, as observed by the presence of cell lysis, an atypical CPE. Subsequently, the pathogenicity of CKK1-1 strains was evaluated in 5-day-old piglets. The CKK1-1 p0 strain was found to have naturally reduced virulence, whereas the CKK1-1 p180 strain showed attenuation, as demonstrated by the 100% survival rate, reduced clinical symptoms, and microscopic changes in tissue analysis. Therefore, the CKK1-1 p180 strain may be useful for the rational design of attenuated vaccines to control emerging PEDV epidemics.

## Figures and Tables

**Figure 1 viruses-16-01108-f001:**
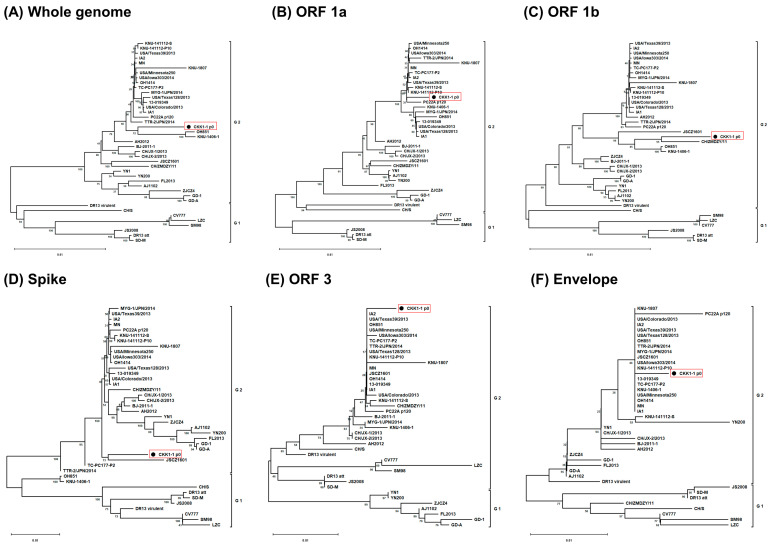
Phylogenetic analysis of the CKK1-1 strain based on the nucleotide sequences of the whole genome, ORF 1a, ORF 1b, spike (S), ORF 3, envelope (E), membrane (M), and nucleocapsid (N) genes. Phylogenetic tree of (**A**) the whole genome; (**B**) ORF 1a gene; (**C**) ORF 1b gene; (**D**) spike gene; (**E**) ORF 3 gene; (**F**) envelope gene; (**G**) membrane gene; and (**H**) nucleocapsid gene. The trees were generated using the neighbor-joining method using the MEGA X software (version 11). Bootstrap analysis was performed with 1000 replicates. The sequence of CKK1-1 p0 is highlighted in red box.

**Figure 2 viruses-16-01108-f002:**
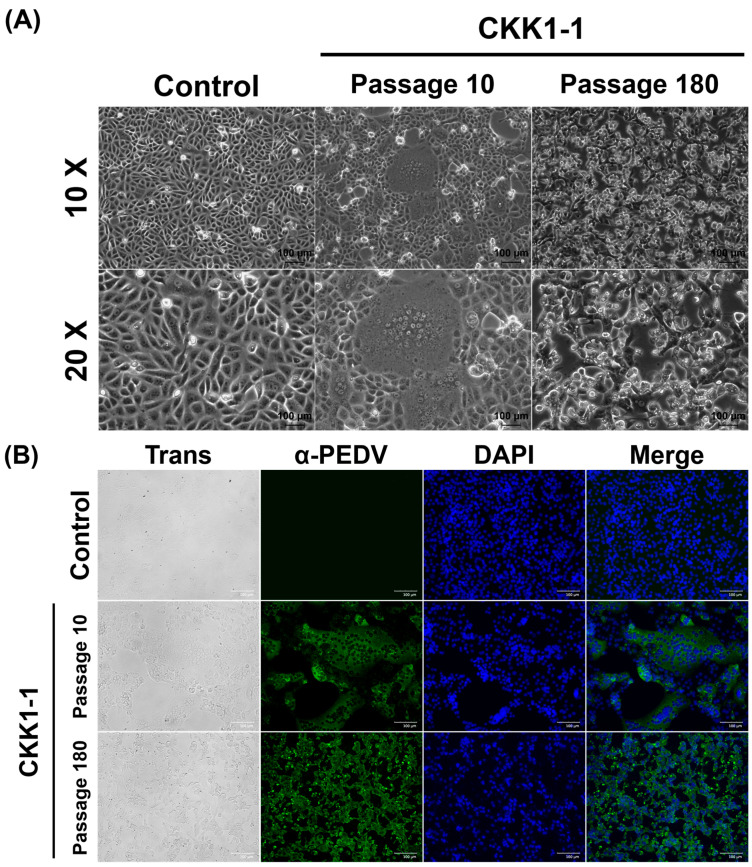
Biological characteristics of the CKK1-1 strain. (**A**) Observation of the cytopathic effects of CKK1-1 at the 10th and 180th passages using microscopy. (**B**) Immunofluorescence assay with PEDV-infected Vero cells using a PEDV-specific monoclonal antibody. The infected cells were fixed (first panels) and incubated with anti-PEDV Mab (second panels). Subsequently, the cells were counterstained with DAPI (third panels) and merged (fourth panels).

**Figure 3 viruses-16-01108-f003:**
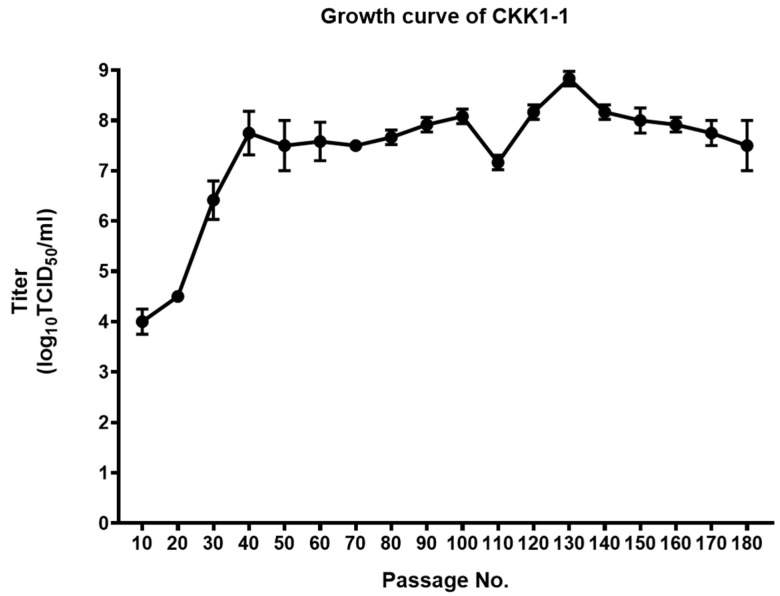
Growth curves of the CKK1-1 strain. Vero cells were infected with 10-fold-diluted cell lysates every 10 passages. The titers were determined using a 50% tissue culture infectious dose (TCID_50_) infectivity assay.

**Figure 4 viruses-16-01108-f004:**
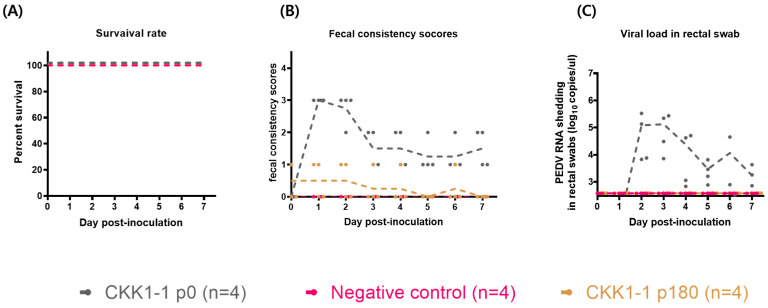
Evaluation of the pathogenicity of the CKK1-1 strain. (**A**) The survival rate of piglets inoculated with either the CKK1-1 strain (passage numbers 0 and 180) or medium (negative control). (**B**) Fecal consistency scores were measured throughout the entire animal experiment according to the standard described in the Materials and Methods section. (**C**) Viral load in rectal swab samples. Quantification of viral shedding was performed by quantitative RT-PCR analysis. Each dot (.) represents an individual sample.

**Figure 5 viruses-16-01108-f005:**
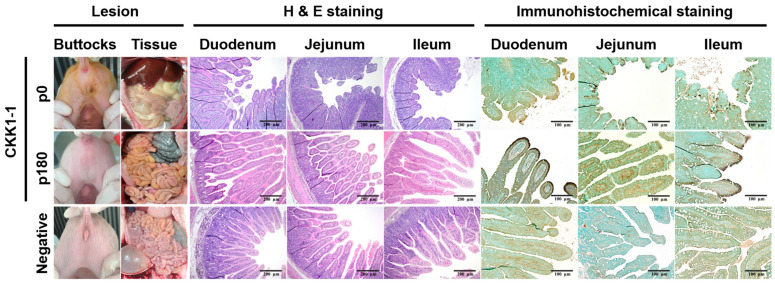
Illustrations of gross lesions identified in clinical symptom, histopathological, and immunohistochemistry analyses. Clinical symptoms with gross lesions were compared among the negative control group and the CKK1-1 p0 and p180 strains. Small-intestine tissue sections were categorized as duodenum, jejunum, and ileum, which were stained with hematoxylin and eosin for histopathological analysis and immunohistochemically stained for PEDV N protein antigen using a monoclonal antibody. The presence of PEDV antigens, indicated by brown staining, was primarily observed in the villous epithelial cells of the CKK1-1 p0 and p180 strains. Images of hematoxylin and eosin staining are shown at 100× magnification, while images of immunohistochemical staining are shown at 200× magnification.

**Table 1 viruses-16-01108-t001:** Summary of the continuous passaging of CKK1-1 under varying culture conditions.

Passage Numbers																																	
Origin	1	2	3	4	… ***	28	29	30	31	…	66	67	68	69	70	71	…	76	77	78	79	80	81	82	83	84	85	…	177	178	179	180	
T15N *	1	2	3	4	…																												
T10N **						1	2	3	4	…	39	40																					
T5N													1	2	3	4	…	9	10														
T2N																				1	2	3	4										
N																								1	2	3	4	…	96	97	98	99	

* TPCK-treated trypsin (T): L-1-tosylamide-2-phenylethyl chloromethyl ketone-treated trypsin. ** Na-GCDCA (N): sodium-glychenodeoxycholic acid; *** omitting numbers.

**Table 2 viruses-16-01108-t002:** Comparison of nucleotide and amino acid sequence identities (%) of different regions between CKK1-1 and other strains.

Identity to CKK1-1/Nucleotide (Amino Acid)										
Strains	Whole Genome	5′ UTR	ORF1a	ORF1b	Spike	ORF3	Envelope	Membrane	Nucleocapsid	3′ UTR
CV777	96.50 (-)	97.14 (-)	96.89 (93.36)	97.48 (98.99)	93.27 (93.09)	96.31 (96.36)	96.38 (98.68)	97.91 (98.21)	95.59 (97.01)	97.21 (-)
CH/S	97.06 (-)	98.95 (-)	97.78 (94.77)	97.92 (98.88)	93.00 (92.93)	97.72 (100)	95.93 (97.33)	97.75 (98.21)	96.27 (97.47)	97.84 (-)
SM98	96.31 (-)	97.15 (-)	96.68 (93.02)	97.38 (98.80)	92.93 (92.34)	96.15 (95.70)	95.47 (95.97)	97.75 (97.30)	95.43 (96.77)	96.88 (-)
DR13_virulent	97.37 (-)	98.95 (-)	97.76 (94.66)	98.21 (99.73)	94.31 (94.17)	98.03 (99.55)	97.76 (100)	98.06 (98.21)	96.90 (97.71)	97.86 (-)
DR13_att	96.97 (-)	98.94 (-)	97.73 (95.14)	97.58 (99.23)	93.19 (92.92)	97.22 (99.03)	96.03 (95.49)	97.61 (97.75)	96.34 (97.71)	98.78 (-)
LZC	96.23 (-)	94.94 (-)	96.75 (93.09)	97.41 (98.84)	92.70 (92.07)	94.90 (94.49)	95.47 (95.97)	96.85 (96.38)	95.35 (96.54)	96.59 (-)
JS2008	97.01 (-)	98.22 (-)	97.77 (95.29)	97.65 (99.15)	93.41 (93.00)	96.88 (98.54)	95.44 (94.59)	97.45 (97.30)	96.34 (97.71)	98.79 (-)
AJ1102	98.38 (-)	99.30 (-)	99.08 (97.97)	98.46 (99.69)	97.18 (97.43)	95.65 (97.28)	98.68 (100)	98.37 (98.21)	96.02 (97.24)	98.79 (-)
GD-1	97.89 (-)	98.95 (-)	98.05 (95.60)	98.45 (99.54)	97.13 (97.21)	94.84 (96.36)	98.23 (98.68)	98.37 (98.21)	96.09 (97.47)	98.48 (-)
ZJCZ4	97.91 (-)	99.30 (-)	97.93 (95.39)	98.33 (99.65)	97.71 (97.73)	95.50 (96.83)	98.68 (100)	98.06 (98.21)	96.26 (97.71)	98.79 (-)
BJ-2011-1	98.74 (-)	98.95 (-)	99.03 (97.88)	98.33 (99.65)	98.41 (98.32)	99.10 (99.55)	98.67 (100)	99.41 (98.66)	99.08 (99.77)	98.48 (-)
SD-M	96.94 (-)	98.94 (-)	97.72 (95.17)	97.58 (99.19)	93.12 (92.53)	97.22 (99.03)	95.93 (95.97)	97.76 (97.75)	96.18 (97.47)	98.48 (-)
AH2012	98.91 (-)	99.30 (-)	99.39 (98.77)	98.54 (99.57)	98.12 (97.81)	98.19 (99.55)	98.67 (100)	99.41 (99.55)	99.08 (99.55)	98.79 (-)
13-019349	99.17 (-)	98.95 (-)	99.56 (98.99)	98.57 (99.65)	98.86 (98.10)	99.55 (100)	99.56 (100)	99.71 (99.55)	99.62 (100)	99.10 (-)
USA/Colorado/2013	99.20 (-)	99.30 (-)	99.58 (99.02)	98.57 (99.65)	98.98 (98.47)	99.40 (99.55)	99.56 (100)	99.71 (99.55)	99.62 (100)	99.10 (-)
YN1	98.60 (-)	99.30 (-)	99.08 (98.00)	98.38 (99.61)	97.76 (97.59)	96.3 (98.20)	99.12 (100)	99.11 (99.11)	98.7 (99.55)	98.48 (-)
USA/Texas128/2013	99.15 (-)	100 (-)	99.56 (98.99)	98.53 (99.54)	98.71 (97.81)	99.55 (100)	99.56 (100)	99.71 (99.55)	99.55 (99.55)	99.08 (-)
IA1	99.18 (-)	99.65 (-)	99.55 (98.96)	98.53 (99.54)	98.96 (98.47)	99.55 (100)	99.56 (100)	99.56 (99.55)	99.62 (100)	99.10 (-)
IA2	99.24 (-)	99.65 (-)	99.74 (99.47)	98.54 (99.65)	98.83 (68.40)	99.55 (100)	99.56 (100)	99.71 (99.55)	99.54 (99.77)	99.10 (-)
MN	99.25 (-)	99.65 (-)	99.76 (99.55)	98.54 (99.61)	98.83 (98.32)	99.55 (100)	99.56 (100)	99.71 (99.55)	99.62 (100)	98.48 (-)
TC_PC177-P2	99.27 (-)	100 (-)	99.73 (99.50)	98.54 (99.61)	98.92 (98.47)	99.55 (100)	99.56 (100)	99.71 (99.55)	99.62 (100)	99.10 (-)
FL2013	98.19 (-)	98.59 (-)	98.79 (97.34)	98.42 (99.57)	96.92 (96.76)	95.5 (96.36)	98.23 (98.68)	98.37 (98.21)	96.02 (98.21)	98.48 (-)
USA/Iowa303/2014	99.25 (-)	100 (-)	99.76 (99.52)	98.54 (99.61)	98.83 (98.47)	99.40 (100)	99.56 (100)	99.71 (99.55)	99.54 (100)	99.10 (-)
USA/Minnesota250	99.24 (-)	100 (-)	99.73 (99.44)	98.54 (99.65)	98.81 (98.40)	99.55 (100)	99.56 (100)	99.71 (99.55)	99.54 (100)	99.10 (-)
OH1414	99.25 (-)	99.65 (-)	99.74 (99.47)	99.57 (99.61)	98.81 (98.47)	99.55 (100)	99.56 (100)	99.71 (99.55)	99.54 (100)	99.10 (-)
YN200	98.46 (-)	99.30 (-)	99.03 (97.83)	98.42 (99.61)	96.79 (96.53)	96.30 (98.19)	97.77 (97.33)	99.26 (98.66)	99.09 (99.55)	98.48 (-)
KNU-141112-P10	99.26 (-)	100 (-)	99.76 (99.58)	98.56 (99.65)	98.81 (98.18)	99.55 (100)	99.56 (100)	99.71 (99.55)	99.62 (100)	99.10 (-)
MYG-1/JPN/2014	99.10 (-)	100 (-)	99.50 (98.85)	98.53 (99.57)	98.69 (97.96)	98.8 (100)	99.56 (100)	99.71 (99.55)	99.62 (100)	98.83 (-)
OH851	98.85 (-)	99.65 (-)	99.36 (98.43)	99.02 (99.73)	96.43 (95.85)	99.55 (100)	99.56 (100)	99.56 (99.55)	99.47 (99.77)	99.10 (-)
TTR-2/JPN/2014	99.14 (-)	99.57 (-)	99.66 (99.33)	98.41 (99.57)	98.76 (98.22)	99.55 (100)	99.56 (100)	99.41 (95.55)	99.32 (100)	98.43 (-)
PC22A_p120	99.15 (-)	100 (-)	99.64 (99.24)	98.49 (99.61)	98.68 (98.02)	99.25 (99.10)	98.67 (98.68)	99.56 (99.11)	99.62 (100)	98.78 (-)
JSCZ1601	98.77 (-)	98.94 (-)	98.92 (97.65)	98.81 (99.65)	98.04 (97.51)	99.55 (100)	99.56 (100)	99.17 (99.55)	98.46 (99.09)	98.48 (-)

**Table 3 viruses-16-01108-t003:** Summary of the results of pathogenicity evaluation.

Groups	Inoculum	Route	No. of Pigs	Mortality Rate [% (No/Total)]	Severe Diarrhea Rate * [% (No/Total)]	Clinical Symptoms	Virus Shedding	Peak Fecal Virus Shedding Titer (log_10_copies/uL); dpi
1	CKK1-1 p0	Oral	4	0 (0/4)	100 (4/4)	Severe	Started at 2 dpi	4.78 ± 0.75, 3
2	Negative control		4	0 (0/4)	0 (0/4)	No diarrhea	N/D **	NA ***
3	CKK1-1 p180		4	0 (0/4)	0 (0/4)	No diarrhea	N/D	NA

* Fecal consistency scores: 0, solid; 1, pasty; 2, semi-liquid; 3, liquid; 4, death. An Fc score of 3 was considered to indicate severe diarrhea; **, no detection; ***, not available.

**Table 4 viruses-16-01108-t004:** Genetic alterations in the CKK1-1 strain depending on the passage number.

Gene	Nucleotide Position	Amino-Acid Position	CKK1-1 Strains	
			P0	P180
ORF 1a/1b	99	33	GAT(D)	GAG(E)
	132	44	TTC(F)	TTT(F)
	833	278	TGC(C)	TTC(F)
	2098	700	TCA(S)	GCA(A)
	2858	953	GTT(V)	GCT(A)
	2941	981	CCC(P)	TCC(S)
	3790	1264	CTG(L)	TTG(L)
	3966	1322	AAT(N)	AAC(N)
	4106	1369	GGT(G)	GAT(D)
	4691	1564	TCT(S)	TTT(F)
	4718	1573	GAC(D)	GCC(A)
	8129	2710	ATC(I)	ACC(T)
	8712	2904	GAG(E)	GAT(D)
	9264	3088	AAT(N)	AAG(K)
	10,301	3434	TCT(S)	TTT(F)
	11,451	3817	AAT(N)	AAC(N)
	13,388	4463	TCA(S)	TTA(L)
	13,979	4660	CTA(L)	CCA(P)
	14,003	4668	AGA(R)	AAA(K)
	14,042	4681	CAG(Q)	CGG(R)
	14,693	4898	CTG(L)	CCG(P)
	16,607	5536	ATG(M)	AGG(R)
	16,609	5537	CTG(L)	TTG(L)
	17,870	5957	ACC(T)	ATC(I)
	18,265	6089	GAT(D)	AAT(N)
	18,311	6104	ACC(T)	ATC(I)
	18,677	6226	TCT(S)	TTT(F)
Spike	79–80	27	TCA(S)	AAA(K)
	397	131	ACA(T)	CCA(P)
	469	157	CAT(H)	TAT(Y)
	974	325	GAC(D)	GCC(A)
	1056	352	TCC(S)	TCT(S)
	1148	383	ACT(T)	ATT(I)
	1571	524	CCT(P)	CAT(H)
	1903	635	CTT(L)	ATT(I)
	1906	636	GAA(E)	CAA(Q)
	1984	662	ATC(I)	GTC(V)
	2148	716	GTT(V)	GTC(V)
	2180	727	AGT(S)	ATT(I)
	2336	779	ACT(T)	AAT(N)
	2675	892	AAA(K)	AGA(R)
	2678	893	AAA(K)	ACA(T)
	2770	924	CGC(R)	AGC(S)
	2894	965	GGT(G)	GCT(A)
	2910	970	GCA(A)	GCG(A)
	2915	972	TTG(L)	TCG(S)
	3153	1051	CAA(Q)	CAC(H)
	3485	1162	GAT(D)	GCT(A)
	3500	1167	ATT(I)	AGT(S)
	3533	1178	ATT(I)	ACT(T)
	3667	1223	ATT(I)	GTT(V)
	3818	1273	AAT(N)	ACT(T)
	3823	1275	ACT(T)	GCT(A)
	3842	1281	GGT(G)	GAT(D)
	4006	1336	GTT(V)	ATT(I)
	4138–4140	1380	ㅡㅡㅡ(ㅡ)	TGA(*)
ORF3	571	191	GAT(D)	TAT(Y)
Envelope	126	42	CAT(H)	CAC(H)
	209	70	CCT(P)	CTT(L)
Membrane	34	12	ATT(I)	GTT(V)
	70	24	ATC(I)	CTC(L)
Nucleocapsid	228	76	TAT(Y)	TAC(Y)
	497	166	AGA(R)	ATA(I)
	551	184	CGT(R)	CCT(P)
	1130	377	GGG(G)	GAG(E)

Insertion (blank) and stop codon indicated by dash (ㅡ) and asterisk (*), respectively.

## Data Availability

Data are contained within the article and Appendix A.

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
