# Peer review of "Genetic and Pathogenic Analysis of a Novel Porcine Epidemic Diarrhea Virus Strain Isolated in the Republic of Korea"

_viruses, 2024, doi:10.3390/v16071108_

Round 1
Reviewer 1 Report
Comments and Suggestions for Authors
The data presented in the manuscript showed that the Korean strain CKK1-1, attenuated through culture conditions with Na-glycochenodeoxycholic acid, has potential as a vaccine candidate. The work is valuable. The search for attenuated PEDV strains is advisable due to the fact that current vaccines do not provide significant protection. The work is interesting but needs major revision. The manuscript is not easy to read. Often the sentences are very colloquial. Especially the results section needs major improvement. Tables and figures also need improvements. The conclusions are consistent with the presented results.
Specific comments:
Introduction
1. citations to text L34-41 are missing
2. L50-52 “Recent studies have reported…..during viral infection”. Citations are missing.
Materials and methods
1. lacks bioethics committee approval to take small intestine samples from the piglets.
2. How do you know that the flocks were infected with PEDV? early confirmatory tests were performed?
3. There is no information on how many samples were tested.
Results
1. text in lines 197-203 has to be rewritten as it is not very understandable.
2. It is incomprehensible to me why there is a quotation in line 203. If this is not the result of analysis then please remove it.
3. The use of the words “other regions” in line 203 is inappropriate. Rephrase the sentence.
4. Rephrase the sentence : “Among the ORFs, the highest genetic variance occurred within the
spike gene, with the highest identity strain being USA/Colorado/2013 (98.98%).” and “The
strain with the lowest identity was LZC (92.70%), which belonged to genogroup 1.” sentences are incomprehensible
5. Remove the sentence “This strain was identified in a 7-day-old piglet with severe diarrhea in the United States [15].” this is not the result of the study
6. The sentence “Through a comparison of genetic information, we demonstrated that the whole genome sequence of the CKK1-1 strain shared 96.23% to 99.27% similarity with the reference strains, with the spike region being the most variable portion among the ORFs.” is a repetition of what was written earlier
7. “Furthermore, CKK1-1 exhibited novel genetic characteristics.” please write what are the new features. This is very important.
8. Line 229. PEDV strains could be classified not clades
9. Remove the sentence “The OH851 strain is widely recognized as a prototype of the U.S INDEL strain [16].” It is not the result of the study.
10. The phylogenetic trees are too small. They cannot be interpreted.
11. “Moreover, the CKK1-1 strain was classified within genogroup 2 in trees based on other ORFs, indicating a novel phylogenetic relationship.” I don't understand. What new relationships?
12. “A prior genomic analysis study showed that the JSCZ1601 strain, isolated in China in 2016, exhibited two amino acid deletions in the spike protein and one insertion in the 5' untranslated region [17].” Remove. It is not the result of this study.
13. “Phylogenetic analysis identified a novel phylogenetic characteristic of the CKK1-1 strain.” please expand the sentence because it is not clear what the authors have in mind.
14. “Additionally, CKK1-1 was affiliated with a new clade within genogroup 2 when compared to the reference strains.” so please divide the strains into subclades to make it more visible.
15. Table 2 is incomprehensible. I suggest removing it.
16. “Initially, PEDV was passaged in virus growth media containing 1 μg/ml TPCK-treated trypsin for up to 27 generations. Subsequently, the trypsin concentration was reduced to 1 μg/ml for 40 generations.” It follows that the concentration of trypsin has not changed.
17. Section 3.3 should be included into the material and methods section.
18. “The isolated PEDV strains were passaged for 180 generations under various conditions (Table 2).” How many PEDV strains?
19. Line 266. “We obtained PEDV-positive small intestine samples from a PED outbreak farm in the Republic of Korea.” How many samples? results are missing.
20. “After genome analysis, we isolated PEDV with novel genetic characteristics, and successfully cultivated it in Vero cells.” Remove “novel genetic characteristics”
21. L280. After 180 passages? Passages were more than 180?
22. Figure 2. please explain the names T15N p10 and N p180.
23. L305. Remove “with novel genetic characteristics”
24. Remove text from lines 325-331. Summaries in each section are unnecessary.
25. Descriptions of tables and figures are sometimes unclear. Not all abbreviations are explained. Please correct.
Discussion
26. L441-474. I suggest shortening this text.
27. L492. “In our animal experiments, we assessed the pathogenicity of the CKK1-1 strain by dividing the groups into CKK1-1 p0, CKK1-1 N_p180, and a negative control.” Change the sentence. Do not use abbreviations.
28. “Conversely, piglets inoculated with the CKK1-1 N_p180 strain and passaged with bile acid for up to 180 passages did not exhibit mortality, and no clinical symptoms were observed through out the experiment.” Rephrase the sentence. From the sentence, it appears that the animals were passaged.
29. “Additionally, the parental strain, which was naturally attenuated, was passaged for up to 180 generations, showing no genetic alterations.” I don’t understand this sentence.
Author Response
Plese see the attachement.

Reviewer 2 Report
Comments and Suggestions for Authors
In the manuscript of Kim et al., the authors describe isolation of new strain of PEDV: CKK1-1. The full genomic sequence was performed, and genome of the strain was compared with genomes of other strains. The virus was passaged 180 times in vitro under various culture conditions reducing concentration of trypsin and presence of sodium-glycochenodeoxycholic-acid. The virus from passage 180 was sequenced again and genetic mutations were identified. Pathogenicity of the virus was evaluated in 5-day-old piglets. Passage 180 virus demonstrated decrease in virulence.
1. Objectives: the objectives and the rationale of the study clearly stated.
2. Methods: The methods are described in sufficient detail to understand the approach used.
3. Results: the results are clearly presented.
4. Interpretation: Conclusions are supported by the obtained data.
5. Other comments:
Line 83 – the virus, not Vero cells, were serially passaged.
Line 199 – what is “low homogeneity”?
Line 219 and 522 – I think words “novel” and “unique” are too strong to describe genetic differences of this strain. I think word “distinct” is more appropriate.
Table 1. You need to explain that numbers in brackets are for amino-acid identities. Consider shortening the table by removing the less significant strains.
Figure 1. There is an error in this figure. Groups 1 and 2 are mislabelled.
Lines 254-256 – “PEDV was passaged in virus growth media containing 1 μg/ml TPCK-treated trypsin for up to 27 generations. Subsequently, the trypsin concentration was reduced to 1 μg/ml for 40 generations.” According to this text, concentration of trypsin was not reduced.
Lines 374-385 – This paragraph has to be moved to the section 3.1.
Round 2
Reviewer 1 Report
Comments and Suggestions for Authors
The authors have made improvements to the manuscript but there are still a few things to improve. Since the manuscript is sometimes hard to read I recommend that a language correction be made.
Specific comments:
1. Line 105 supernatant was filtered instead supernatants were filtered
2. Line 106 Remove “that tested positive by RT-PCR” since only one supernatant was tested
3. Rephrase the sentence “The nucleotide identity of the whole genome ranged from 96.23% to 99.27%” I propose : “The nucleotide identity of the entire genome of all PEDV strains analyzed ranged from 96.23% to 99.27%.”
4. Lines 217-218. “The highest nucleotide identity observed across the entire genome was 99.27%, found in the TC_PC177-P2 strain., The primary distinction of the TC_PC177-P2 strain was a significant deletion in the S protein, comprising 197 amino acids.” The sentence has to be corrected. It needs to be written that this is a comparison of the CKK1 strain with TC_PC177-P2. At least that's how I understand it. Unfortunately, readers have to guess what it is about.
5. “Among the ORFs, the highest genetic variance occurred within the spike gene, with the strain USA/Colorado/2013 showing the highest identity at 98.98%. The strain with the lowest identity was LZC (92.70 %), belonging to genogroup 1.” Unfortunately, the sentences have not been corrected. It is still unclear which sequences of which strains are being compared.
Comments on the Quality of English LanguageI recommend stylistic editing by a native speaker.
